# Agnostics: Learning to Synthesize Code in Any Programming Language with a Universal Reinforcement Learning Environment

**Aleksander Boruch-Gruszecki,**[1,*] **Yangtian Zi,**[1] **Zixuan Wu,**[1] **Tejas Oberoi,**[2]
**Carolyn Jane Anderson,**[3] **Joydeep Biswas,**[2] **Arjun Guha**[1]
[1]Northeastern University, [2]University of Texas, [3]Wellesley College
[*]research@abgru.me

## Abstract

Large language models (LLMs) excel at writing code in *high-resource* languages such as Python and JavaScript, yet stumble on *low-resource* languages that remain essential to science and engineering. Besides the obvious shortage of pre-training data, post-training itself is a bottleneck: every new language seems to require new datasets, test harnesses, and reinforcement-learning (RL) infrastructure.

We introduce Agnostics, a *language-agnostic* post-training pipeline that eliminates this per-language engineering. The key idea is to judge code solely by its externally observable behavior, so a single verifier can test solutions written in *any* language. Concretely, we (i) use an LLM to rewrite existing unit-test datasets into an I/O format, (ii) supply a short configuration that tells the verifier how to compile and run a target language, and (iii) apply reinforcement learning with verifiable rewards (RLVR) in a robust code execution environment.

Applied to five low-resource languages—Lua, Julia, R, OCaml, and Fortran—Agnostics (1) improves Qwen-3 4B to performance rivaling other 16B–70B open-weight models; (2) scales to larger and diverse model families (Qwen-3 8B, DeepSeek Coder 6.7B Instruct, SmolLM 3, Phi 4 Mini); and (3) for ≤16B parameter models, sets new state-of-the-art pass@1 results on MultiPL-E and a new multi-language version of LiveCodeBench which we introduce.

We release the language-agnostic training datasets (Ag-MBPP-X, Ag-Codeforces-X, Ag-LiveCodeBench-X), training code, and ready-to-use configurations, making RL training for *any* programming language as easy as a few lines of YAML.

## 1 Introduction

Large language models (LLMs) are remarkably good at programming tasks, especially when coding in *high-resource programming languages* such as Python and JavaScript. Their proficiency in *low-resource programming languages*, such as Fortran, Julia, and others, is far more limited. This gap appears both on benchmarks (Cassano et al., 2023) and in popular discourse. Many low-resource languages are adapted to and widely used in particular sectors such as computational science (e.g., Julia, Fortran), medicine (e.g., Mumps), data science (e.g., R), and others. Methods for improving LLMs on such languages would help programmers in these sectors truly take advantage of LLMs.

The capability gap between high-resource and low-resource programming languages occurs for two reasons. First, there is *vastly* more training data for some languages. For example, The Stack V2 (Lozhkov et al., 2024a), the largest public training corpus of code, has ≈200GB of Python but only ≈2GB of Julia and Fortran. Thus pretraining on code makes models significantly better at Python. A subtler reason is the availability of post-training datasets and techniques. Contemporary LLMs are developed with an extensive post-training process that relies on (a) high-quality curated data for supervised fine-tuning, and (b) carefully designed environments for reinforcement learning, which must be able to execute and verify model-generated solutions. Both of these require significant human expertise, which is hard to find for low-resource programming languages.

Our goal in this work is to facilitate post-training LLMs on low-resource programming languages, working towards closing the resource gap. Our key idea is that for a large class of programming tasks, correctness can be stated as a property not of functions or code snippets, but of the entire program's observable behavior (e.g., I/O). Furthermore, if its correctness can be tested with a *verifier* program, such a *verifier* with appropriate problems and test cases can be used to make a universal reinforcement learning environment which can be instantiated for nearly any programming language. In fact, the verifier's implementation language is independent from the one being learned. This approach matches the formulation of some existing post-training datasets (even if they are intended for Python/C++), and we can reformulate language-specific datasets into this format with LLMs.

Our approach, Agnostics, works based on this insight as follows (Figure 1). 1) We use an LLM to reformulate language-specific datasets into our uniform language-agnostic format. 2) To target a particular language, we generate prompts and instantiate the verifier based on a small (4-5 line) configuration file. 3) We apply reinforcement learning with verified rewards (RLVR) using a robust, language-agnostic execution sandbox that we develop. 4) The result is a model specialized to the target language. Agnostics particularly excels at finetuning models for low-resource languages, as it does not rely on high-quality datasets specific to a particular language.

### Contributions

1. Agnostics, a post-training pipeline for coding in arbitrary programming languages;
2. The best-performing open-weights ≤16B models for Lua, R, Julia, OCaml and Fortran;
3. Three Agnostics datasets: Ag-MBPP-X, Ag-Codeforces-X, and Ag-LiveCodeBench-X, based on MBPP (Austin et al., 2021), Open-R1 Codeforces (Penedo et al., 2025) and LiveCodeBench (Jain et al., 2024b) respectively.
4. A small and carefully designed Agnostics training framework, including a parallel code execution sandbox, sampling, rewards computation, GRPO, and model back-propagation.

Our leaderboard, data, code, and artifacts are referenced at agnostics.abgru.me and in §5.

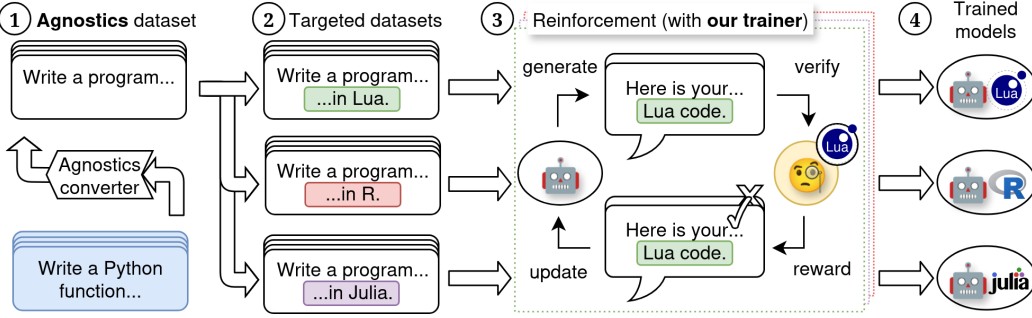

Figure 1: **Overview: Agnostics Data Preparation and Training. (1)** We reformulate existing coding datasets to our format. **(2)** We adapt the language-agnostic datasets to a particular programming language. **(3, 4)** We reinforce coding via Group Relative Policy Optimization (Shao et al., 2024; DeepSeek-AI et al., 2025), verifying the programs in our code execution sandbox.

## 2 BACKGROUND AND RELATED WORK

**Data Scarcity for Low-Resource Languages** General-purpose LLMs have been pretrained on code for several years, both because LLMs are widely used for practical programming tasks, and because pretraining on code improves their general reasoning abilities (Ma et al., 2023). Code-specialized models may be trained from a general-purpose model checkpoint or exclusively on code (e.g., Xu et al. (2022); Allal et al. (2023)).

| Language | % | Language | % |
|---|---|---|---|
| JavaScript | 17.04 % | Lua | 0.53 % |
| Java | 8.38 % | R | 0.35 % |
| C++ | 5.41 % | Fortran | 0.07 % |
| Python | 3.56 % | Julia | 0.10 % |

Figure 2: High-resource (left) and low-resource (right) languages in the Stack v2.

However, the publicly available pretraining data for code is heavily skewed toward a handful of programming languages. E.g., consider The Stack V2 (Lozhkov et al., 2024b), the largest public code pretraining dataset, with code from GitHub and dozens of other sources. The Stack V2 is dominated by relatively few programming languages: just 10 of 619 languages account for over 90% of the dataset. We want to develop models for low-resource programming languages that each account for $\leq 0.5\%$ of publicly available code (Figure 2). We can imagine working around this data scarcity in a few ways. However, previous work shows that up-sampling low-resource languages during pretraining only leads to small benchmark improvements (Orlanski et al., 2023), while fine-tuning on the pre-training data for low-resource languages has negligible impact (Cassano et al., 2024). Thus, it is not clear how to make further gains from existing natural data.

**Synthetic Data for Low-Resource Languages**   If natural data is not available for a task, it is possible to use LLMs to generate synthetic fine-tuning data (Wang et al., 2023), and there are many techniques for building code-centric supervised fine-tuning datasets (e.g. Luo et al. (2023); Wei et al. (2024c;b)) that work remarkably for Python. Although these approaches could in principle be applied to any programming language, Cassano et al. (2024) show that without distillation or verification (e.g., see Hu et al. (2025); Wei et al. (2024a)), synthetic tasks and code for low-resource programming languages are low-quality, and models fine-tuned on them perform poorly.

MultiPL-T (Cassano et al., 2024), similar to TransCoder-ST (Roziere et al., 2021) and CMTrans (Xie et al., 2024), couples synthetic data generation with verification using rejection sampling: it generates up $n$ candidate programs in a target, low-resource language and only fine-tune models on generations that pass hidden unit tests that it translates from Python. However, the MultiPL-T approach has two significant limitations. (1) For the verifier to not reject all samples, the model must be able to generate a working program within $n$ attempts. In MultiPL-T, $\approx 30\%$ of prompts produce a working program for $n \in \{50, 100\}$ attempts, and the rest are discarded. We train on much harder problems, and estimate that rejection sampling would require an order of magnitude more resources for a comparable acceptance rate (§4.3). (2) For each low-resource language of interest, MultiPL-T requires writing a little compiler to translate test cases and function signatures from Python to the target language. MultiPL-T only supports a limited set of built-in Python types (e.g., no classes) and dictates that all Python types and values must faithfully map to the target language. However, depending on the problem and language, the natural data representation may not map cleanly to Python. This can lead to peculiar, unidiomatic translations that require deep language expertise to get right. The Agnostics approach is far easier to use than MultiPL-T, and only requires the user to know how to compile and run a program in the target language from the shell.

**Reinforcement Learning on Coding Tasks**   DeepSeek R1 (DeepSeek-AI et al., 2025) popularized RL on LLMs with rule-based rewards, instead of learned reward models. R1 reports applying RL to coding tasks without further dataset details. A number of papers apply RL to the NL to code task (Zeng et al., 2025; Gehring et al., 2024; Jain et al., 2025). These techniques target Python and show that RL can improve LLM capabilities beyond what supervised fine-tuning allows alone.

However, the key benefit of RL is that it can train a model to do tasks for which high-quality supervised fine-tuning data is unavailable. There are recent examples of using RL for code optimization (Du et al., 2025; Nichols et al., 2024), resolve GitHub, issue resolution (Wei et al., 2025), and iterative development (Zhou et al., 2025). These papers target tasks in high-resource languages (C++, Java, and Python) whereas Agnostics targets several low-resource languages.

## 3   THE AGNOSTICS APPROACH

Our approach comprises (1) a *data preparation* stage which reformulates language-specific programming tasks to be language-agnostic, and retargets language-agnostic datasets to a programming language of interest (1, 2 in Figure 1); and (2) the *training* stage which uses the GRPO algorithm and an efficient, language-agnostic verification framework (3, 4 in Figure 1). Our tasks ask for programs by describing their behavior. The test cases are samples of this behavior, and a verifier program can check if a solution behaves according to the sample. In this paper, we limited ourselves to working with tasks asking for programs which read data from the standard input, compute a unique answer, and write it to the standard output. Hence, the datasets we prepared share one verifier.

```
# Write a python function to
# identify non-prime numbers.
def is_not_prime(n):
    ...
```

```
assert is_not_prime(2) == False
assert is_not_prime(10) == True
```

**Instruction:** Given an integer $N$ ($N \geq 2$), determine if it is a non-prime number. Output 'True' if the number is non-prime, 'False' otherwise. Input format: a single integer $N$ ($N \geq 2$). Output format: a single line containing 'True' or 'False'.

| Input | Output |
|-------|--------|
| 2 | False |
| 10 | True |

(a) An MBPP task prompt and associated tests (in gray).  (b) The task and tests reformulated for Agnostics.

Figure 3: For dataset preparation, we use an LLM to reformulate fine-tuning datasets with language-specific prompts and tests (above) into equivalent language-agnostic programming tasks.

## 3.1 DATASET PREPARATION

Some datasets, like Open-R1 Codeforces (Penedo et al., 2025), already define tasks in the desired I/O style. More commonly, however, code datasets provide a set of unit tests. Figure 3a shows a representative item from MBPP: it has a natural language problem description and a Python function signature that comprise the prompt, and a suite of tests used to test model-generated code. These datasets can be easily translated into the I/O format.

To make such problems language-agnostic and compatible with our verifier, we prompt an LLM to reformulate each task so that the program communicates exclusively via plain-text standard in and standard out. We ask the model to spell out concrete I/O conventions—number of decimal places, newline versus comma separators, ordering of values, and so on—so that the expected behavior is unambiguous. Figure 3b shows the reformulated example. §A presents the prompt we use to reformulate MBPP; other datasets might require small changes to the prompt.

## 3.2 PROGRAMMING LANGUAGE PREPARATION

To prepare a new language, we author a small configuration file with two purposes. First, it defines a *prompt prefix* (prepended to each problem by the trainer) which instructs the model to produce code in the target language. Second, the configuration file specifies the shell commands to install the language toolchain and run code. In our experience, a prompt prefix simply asking for a solution in language $L$ is enough for more widespread languages with $\geq 5\%$ base accuracy. However, when starting from near-zero accuracy, a longer prefix can help prevent common mistakes. E.g., our R language configuration (Figure 4) features a longer prompt explaining the quirks of I/O APIs in R.[1]

If a model barely knows a programming language, a good prefix can help it. Still, writing the prefix takes manual effort. For OCaml and Fortran, we let a base model generate several faulty snippets, and asked a capable LLM (OpenAI o3) for advice based on the snippets with the following prompt.

> *What follows are several Fortran programs. You'll see that most of them are wrong. Read them carefully and identify the Fortran programming mistakes that I'm making. Ignore algorithmic mistakes, and focus on my misconceptions about Fortran. Come up with advice on how I should program Fortran correctly. Distill this advice into 10-20 sentences.*

We use the resulting instructions verbatim (see §B) when training models. The prefix only needed to slightly raise the model's train split performance; base accuracy as low as $0.09\%$ was enough for the model to start learning (see §4.3). Configuring the two languages took 1 hour each.

## 3.3 TRAINER AND CODE EXECUTION

The Agnostics trainer uses the Group-Relative Policy Optimization (GRPO) reinforcement learning algorithm (Shao et al., 2024), with verifiable rewards (DeepSeek-AI et al., 2025), and further

---

[1]There are 3 ways to run R, 3 I/O APIs, and only one portable way to read from standard in.

```
install: apt-get install -y r-cran-tidyverse
filename: snippet.R
execute: Rscript snippet.R
prompt: |
  Use R version 4. Use `readLines(con = file("stdin"))` to read from
  stdin. Use the `n` argument to read the first `n` lines. For example:
  ```r
  input <- readLines(con = file("stdin"), n = 1)
  n <- as.integer(input)
  cat(n) # print the first line of input
  ```
  Also, please remember to use `cat` to print output.
```

Figure 4: An Agnostics configuration snippet for R (slightly rephrased for presentation).

common tweaks to improve its efficiency (Yu et al., 2025). We couple the algorithm with a language-agnostic code execution framework designed to be robust and efficient.

**Trainer**  The trainer instantiates the GRPO algorithm as follows. Let $(x, \{(in_k, out_k)\}_{k=1}^K) \sim \mathcal{D}$ be a dataset of language-agnostics tasks, where $x$ is the task prompt and $\{(in_k, out_k)\}_{k=1}^K$ is the set of I/O examples. Let $P$ be $L.\texttt{prompt}$ from a language configuration $L$ (e.g., Figure 4). From the behavior policy $\pi_{\theta_{\text{old}}}$ we sample a group $G$ of candidate responses $\{y_i\}_{i=1}^G \sim \pi_{\theta_{\text{old}}}(\cdot \mid P, x)$. We assign each candidate a reward $R_i$, with $R_i = 1$ if the execution environment (described later) verifies that the extracted program behaves as in the I/O examples $(in_k, out_k)$ and $R_i = 0$ otherwise. We turn group rewards into sequence-level advantages $\hat{A}_i$, and update the policy with the objective

$$\mathcal{L}_{\text{GRPO}}(\theta) = \mathbb{E}_{\{(x,\_)\sim\mathcal{D}, \{y_i\}_{i=1}^G \sim \pi_{\theta_{\text{old}}}(\cdot|P,x)\}}$$

$$\left[ \frac{1}{G} \sum_{i=1}^G \frac{1}{|y_i|} \sum_{t=1}^{|y_i|} \min\Big( \text{clip}\big(r_{i,t}(\theta), 1-\varepsilon, 1+\varepsilon\big) \hat{A}_i, r_{i,t}(\theta) \hat{A}_i \Big) \right],$$

$$\text{where} \qquad r_{i,t}(\theta) = \frac{\pi_\theta(y_{i,t} \mid P, x, y_{i,<t})}{\pi_{\theta_{\text{old}}}(y_{i,t} \mid P, x, y_{i,<t})}, \qquad \hat{A}_i = \frac{R_i - \text{mean}(\{R_j\}_{j=1}^G)}{\text{std}(\{R_j\}_{j=1}^G)}.$$

We omit the KL-divergence term, similar to Yu et al. (2025). We also considered and decided against a reward for a partially-correct answer (see §C.3). We tried to reward the model for code which runs without errors but produces wrong output or for code which only passes the public tests (if there are any). In both cases the models were very likely to learn how to exploit the reward, e.g., by producing empty programs or by hard-coding the public tests (and claiming to produce a "draft answer").

**Code Execution**  Our verifier, a language-agnostic code execution sandbox, (1) extracts a program from each candidate; (2) compiles it if needed; and (3) tests it on the I/O examples $\{(in_k, out_k)\}_{k=1}^K$.

To extract the code, we instruct the model to put it in a Markdown block, which all major instruction-tuned models do by default. Since we rely on the native format of the model, we do not need to train the model with a format reward. This guarantees that the increases in rewards we see during training are real improvements and not merely the result of the model learning to format correctly.

For each language, we build and cache an OCI (2025) container using the configuration $L$. To build the container, we install the language compiler and runtime (the script $L.\texttt{install}$), and include a generic execution harness which runs and tests candidate programs. The execution harness runs continuously in the container, waiting for triples with the candidate program, the set of input/output examples, and timeouts. The harness (1) writes the program to disk (to $L.\texttt{filename}$), (2) compiles it if needed ($L.\texttt{compile}$), (3) runs it on each received input ($L.\texttt{execute}$), and verified that it produces the expected output. The harness imposes timeouts on the compilation step and each execution, and returns reward $0$ on any timeout or failed verification. It is important to have timeouts for both compilation and execution. This prevents pathologies such as unbounded macro expansion in Julia (caught by the compile timeout) and infinite loops (caught by the execution timeout). Using containers also allows us to limit CPU, memory, and filesystem usage; no elevated privileges are

granted to the generated program. Although the current datasets only specify tasks by standard I/O, the same sandbox can safely accommodate problems that read/write from network or disk.

A subtle resource limit that we impose is a limit on the size of output. Even with a short timeout such as 30 seconds, a pathological candidate program can output tens of gigabytes of text. This can crash the verifier if it naïvely tries to read and store all output. Instead, the verifier maintains a fixed-size (5MB) read buffer and immediately kills programs which overflow it.

Overall, this design lets us keep a pool of warm containers for the duration of training: we find that spawning a fresh container is two orders of magnitude slower than re-using an existing one. In our experiments, a single training run may involve testing 150,000 programs, each on several I/O examples. Most of the generated programs are faulty and some behave badly, e.g., they either timeout, consume too much memory, or produce too much output. So containers do occasionally crash or need to be killed, and our execution environment handles this automatically.

Finally, to improve compile times, our execution environment mounts a RAM disk in each container. Compilation may be slow due to creating many intermediate files, and indeed some large C++ projects, e.g., Firefox, recommend using a RAM disk to speed up their builds (Firefox, 2025).

**Implementation**  We implement the trainer and execution environment with Ray (Moritz et al., 2018), which facilitates multiprocessing and distributed computing. In particular, Ray lets us distribute the training over a network of heterogeneous nodes, which allows running the trainer on a node specialized for GPU work and the execution environment on a node specialized for CPU work. Ray also lets us easily separate group generation and loss computation into inter-communicating processes. Running the two in parallel significantly speeds up training, as we found that they take a roughly comparable amount of time. The execution environment is also an actor and manages containers with Python `asyncio` coroutines, not actors, to minimize inter-process data copying.

### HYPERPARAMETERS

We use the AdamW optimizer (Loshchilov & Hutter, 2019) with a learning rate of $5 \times 10^{-6}$ and a cosine decay schedule with a warmup of 0.1 epochs. We process 4 prompts in each batch, with group size 32 per prompt. When training we use temperature 0.7 and disable reasoning in hybrid models. (Many generations still show reasoning-like text, either before the answer or in comments.) We describe how we chose the hyperparameter values and how we evaluted different values in §C.1.

## 4  EVALUATION

To evaluate Agnostics, we train and benchmark models on 5 low-resource programming languages. We measure pass@1 accuracy with reasoning disabled, 20 samples per prompt at temperature 0.2. We trained the models for 1 epoch, unless specified otherwise.

### 4.1  TRAINING DATASETS

**Ag-Codeforces-X**, the main dataset we use for training, was created based on competitive programming problems from the Open-R1 Codeforces dataset (Penedo et al., 2025). Few adjustments were necessary, since the problems already specified programs and tests using standard I/O. The train split contains 5369 problems.

**Ag-MBPP-X**, the other training dataset we use, was created from MBPP as explained in §3.1.

Both datasets were analyzed to ensure no contamination with benchmarks we use. See §A for a more detailed description of how the datasets were prepared.

### 4.2  BENCHMARKS

We evaluate Agnostics with the following benchmarks.

**MultiPL-E** (Cassano et al., 2023) is a well-established benchmark, frequently used to evaluate the performance of new LLMs on a broad set of languages (e.g., Kimi Team (2025); Yang et al. (2025); Grattafiori et al. (2024); ByteDance et al. (2025)). MultiPL-E was prepared by compiling

HumanEval (Chen et al., 2021) prompts and unit tests from Python to each target language. Each MultiPL-E programming language requires a ≈500 LOC prompt and test translator, considerably more effort than writing an Agnostics configuration file. A major limitation of MultiPL-E is being too easy for frontier models. With Python, frontier models are now evaluated on solving programming contest problems (Jain et al., 2024b); no multi-language benchmarks are as challenging.

**Ag-LiveCodeBench-X**, a contribution of this paper, is a new multi-language benchmark derived from LiveCodeBench. (The benchmark has no overlap with our training data; see §A.) Live-CodeBench 5.0 has 880 problems; 381 problems have Python starter code and test cases, while the remaining 499 problems instead use standard I/O to specify and test solutions. We used the latter problems to transform LiveCodeBench into an Agnostics dataset. Accordingly, benchmarking a new programming language with Ag-LiveCodeBench-X is straightforward: we can reuse our trainer's language configurations and execution environment (§3.3). Our results and leaderboard (see §1) show that Ag-LiveCodeBench-X is far harder than MultiPL-E and is challenging for frontier models as of February 2026.

## 4.3 RESULTS

We now present our results. We use a few abbreviations in the tables. Ag-LCB-X stands for Ag-LiveCodeBench-X; we clarify abbreviated model names in the text. Highlighted rows present our models; note that each cell in such a row presents the score of a *different* model trained on programming language X. We compute the score as explained in §4.

**SOTA small LLMs for low-resource PLs** Using Agnostics, we train Qwen 3 4B and 8B on Ag-Codeforces-X specialized to Fortran, Julia, R, Lua, and OCaml. The resulting models are state-of-the-art at coding in low-resource programming languages among open-weight models with ≤16B parameters.

Agnostics training yields significant improvements on Ag-LiveCodeBench-X (tables 1 and 2). (i) On every language, our models match or outperform DeepSeek Coder v2 Lite Instruct (16B), and their performance comes close to or exceeds that of Qwen 3 32B and Llama 3.2 70B. (ii) The OCaml and Fortran scores improve from near zero to ≈7% and ≈15%, outperforming even some frontier models. Importantly, these scores show what the models learned from training: during evaluation we omit the prompt prefix used to facilitate learning (§3.1). (iii) Finally, the pass@1 scores improve by a factor of 1.5–2x over the base model. This indicates that Agnostics can improve models beyond usual training on all the publicly available code data, as we can safely assume that the Qwen models were trained on all such data, like the Llama models (Grattafiori et al., 2024).

Models trained with our approach generalize over the competitive programming format: the improvements are not limited to synthesizing programs using standard I/O. To demonstrate this, we evaluate them on the established MultiPL-E benchmark. It features problems which ask for Python functions operating on usual Python data structures, and we find that our training also significantly improves the models on such problems (Table 3).[2] We also confirmed that our training does not lower performance on other programming languages (§C.4).

| Model | Ag-LCB-X | | |
| X= | Lua | Julia | R |
|---|---|---|---|
| Llama 3.3 70B Ins | **25** | 22 | 13 |
| Qwen 3 32B | 22 | **26** | 17 |
| DSC v2 Lite Ins 16B | 13 | 12 | 9 |
| Qwen 3 4B | 11 | 10 | 10 |
| Qwen 3 8B | 11 | 9 | 9 |
| Qwen3-4B-MBPP-X | 15 | 15 | 9 |
| Qwen3-4B-CF-X | 23 | 22 | 15 |
| Qwen3-8B-CF-X | **25** | 25 | **19** |

Table 1: Ag-LCB-X pass@1.

| Model | Ag-LCB-X | |
| X= | OCaml | Fortran |
|---|---|---|
| *Sonnet 4* | 6 | 6 |
| Llama 3.3 70B Ins | **7** | 3 |
| Qwen 3 32B | 2 | 1 |
| DSC v2 Lite Ins 16B | **7** | 6 |
| Qwen 3 4B | 1 | 0 |
| Qwen 3 8B | 0 | 0 |
| Qwen3-4B-CF-X | **7** | 15 |
| Qwen3-8B-CF-X | **7** | **17** |

Table 2: Ag-LCB-X pass@1.

| Model | MultiPL-E | | |
| X= | Lua | Julia | R |
|---|---|---|---|
| Qwen 3 4B | 61 | 51 | 36 |
| Qwen 3 8B | 63 | 53 | 44 |
| Qwen3-4B-MBPP-X | 51 | **62** | 41 |
| Qwen3-4B-CF-X | 64 | 54 | 43 |
| Qwen3-8B-CF-X | **68** | 61 | **52** |

Table 3: MultiPL-E pass@1.

---

[2]Note that MultiPL-E does not support OCaml and Fortran.

Figure 5 shows the GRPO batch pass@1 rates seen when training Qwen3-4B-CF-X. All the models follow similar curves, partially due to being trained on the same data permutation. Nearly all the models slowly keep improving almost until the dataset end. We also observed the train and test split rewards to be correlated with each other (§C.2).

**Agnostics scales to larger models**   To test if the gains from Agnostics training scale with model size, we train the Qwen 3 8B model on Ag-Codeforces-X specialized to Lua, Julia and R and benchmarked it on Ag-LiveCodeBench-X and MultiPL-E (tables 1 and 3). The Qwen3-8B-CF-X models show significant gains on both benchmarks, improving over their 4B counterparts. We expect Agnostics to scale to even larger models, with appropriate computing resources. However, we found that Agnostics training on Ag-Codeforces-X does *not* improve two smaller models, Qwen 3 1.7B and Llama 3.2 3B Instruct, perhaps due to the problems being too difficult for models of this size.

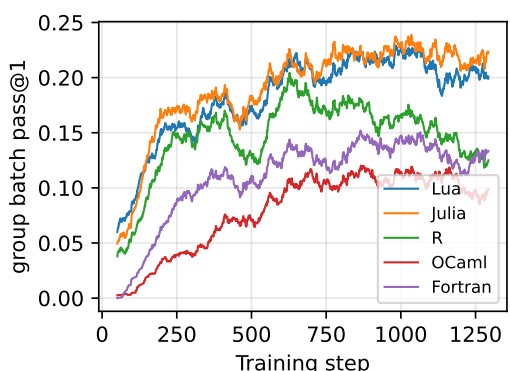

Figure 5: Group batch pass@1, Qwen3-4B-CF-X.

**Agnostics works with easier problems**   All of the models we discussed so far were trained on Ag-Codeforces-X. To show that Agnostics works with other datasets, we also train models for Julia, Lua, and R using the MBPP training set. (§3.1 describes how we prepare MBPP.) MBPP problems are trivial compared to the Open-R1 Codeforces problems (see Figure 3a), and we cannot expect models trained on the MBPP problems to be as good as ones we presented before. Still, training on MBPP improves Lua and Julia performance (tables 1 and 3). The table shows a small drop in R performance on Ag-LiveCodeBench-X, but a significant improvement on MultiPL-E.

**Agnostics works on multiple model families**   To show that Agnostics works on non-Qwen models, we train SmolLM 3 (SmolLM3 Team, 2025), Phi 4 Mini Instruct (Microsoft, 2025), and DeepSeek Coder 6.7B Instruct (Guo et al., 2024) on Ag-Codeforces-X specialized to Lua, Julia, and R. Agnostics improves these models' performance on all languages, as measured by MultiPL-E and Ag-LiveCodeBench-X (Table 4).

| Model | MultiPL-E | | | Ag-LCB-X | | |
|---|---|---|---|---|---|---|
| X= | Lua | Julia | R | Lua | Julia | R |
| SmolLM3 3B | 11 | 12 | 18 | 1 | 2 | 2 |
| Phi 4 mini ins | 40 | 39 | 34 | 8 | 8 | 5 |
| DSC 6.7B Ins | 40 | 54 | 37 | 8 | 5 | 8 |
| SmolLM3-3B-CF-X | 14 | 14 | 21 | 8 | 8 | 6 |
| Phi4-mini-ins-CF-X | 41 | 43 | 35 | **12** | 8 | **12** |
| DSC-6.7B-Ins-CF-X | **42** | **55** | **52** | 9 | **9** | 10 |

Table 4: Non-Qwen models, pass@1.

Note that DeepSeek Coder 6.7B is a relatively old LLM, superseded by the much larger DeepSeekV2 and V3 models. Unlike Qwen 3, DeepSeek Coder is not trained with reinforcement learning, but is only an instruction-tuned model. Thus this result also shows that Agnostics can work on models that have had relatively limited post-training.

**Agnostics outperforms distillation**   So far we discussed training a model on its generations. An alternative is to distill a larger model (assuming one exists). As larger models do not perform very well on many low-resource programming languages, one can expect distillation to be less effective.

We run the following experiment to verify this claim. Using Sonnet 4 Thinking, we synthesize Fortran solutions to Ag-Codeforces-X problems, creating a training set of 1,987 items. (For 13 items, Sonnet 4 (sonnet-4-20250514) with extended thinking does not produce a response within its reasoning budget.) To make sure generating the training items does not use significantly more compute compared to Agnostics training, we use at most 32K reasoning tokens, spending $96 to generate the items. We fine-tune Qwen 3 4B for 3 epochs (batch

| Model | Ag-LCB-Fortran |
|---|---|
| Sonnet 4 Thinking (teacher) | 12 |
| Qwen 3 4B (student) | 0 |
| 1 epoch | 3 |
| 2 epochs | 3 |
| 3 epochs | 2 |

Table 5: Distillation experiment results.

size 64, learning rate $2 \times 10^{-5}$, cosine learning rate decay with warmup ratio 0.1). Table 5 shows the resulting models reach scores far lower than the 15% of Qwen3-4B-CF-Fortran (Table 2).

**Agnostics outperforms rejection sampling** We also ran a small experiment to confirm that rejection sampling would be prohibitively expensive in our case. In rejection sampling with supervised fine-tuning , we prompt the model to synthesize $n$ solutions to each task, reject solutions that fail tests, and fine-tune on the task-solution pairs that pass tests. Cassano et al. (2024) use this approach to get solutions for ≈30% of the tasks in their dataset.

The efficiency of this approach depends on the hardness of the task and the capabilities of the model. In this paper, we use newer models that are marginally better at low-resource languages (based on MultiPL-E benchmark results). The task of Cassano et al. (2024) is to translate a simple, self-contained Python function from the model's pretraining data into an equivalent function in another programming language. This task is significantly easier than the Ag-Codeforces-X task, which is to solve a competitive programming problem in a low-resource language without any reference code.

Rejection sampling would be prohibitively expensive for the low-resource programming languages we consider. Cassano et al. (2024) report a 30% success rate on their code translation task. During Agnostics training, Qwen3-4B-CF-Fortran generated a correct answer to a train split problem only 6.64% of the time, generating 11400 verified programs overall. We also sampled responses to the same problems from the base model of Qwen3-4B-CF-Fortran, Qwen 3 4B, taking the same amount of samples with the same generation parameters as used during training. The base model succeeded 0.09% of the time, generating only 158 test-passing programs.

## 4.4 QUALITATIVE IMPROVEMENTS

We now take a deeper look at how training with Agnostics qualitatively improves Qwen 3 4B. First, we define a taxonomy of common bugs; we prompt o3 to classify bugs in a sample of faulty programs and lightly edit the suggestion. §D has the full taxonomy and the prompt used to develop it. The taxonomy spans fundamental programming errors such as syntax errors, and subtler mistakes such as logic flaws.

We sample 100 problems from Ag-LiveCodeBench-X, and for each take five OCaml programs produced by Qwen 3 4B and the models trained on Ag-Codeforces-X. Recall from Table 2 that our models significantly outperform the base Qwen 3 models on our benchmark. With Sonnet 4, we use the taxonomy to classify the bugs in each program.

Figure 6 shows the bug distribution for OCaml. We see that the base Qwen 3 4B model makes many more fundamental OCaml programming mistakes. More programs have syntax errors (55% vs 35% after training), more programs misuse builtin functions (60% vs 32%), and so on. We do observe a small increase in logic flaws (18% vs 25% after training). When a program is full of syntax errors and hallucinated functions, it is hard to decide if the algorithmic approach is correct; training eliminates these shallow bugs and lets deeper issues manifest. Models trained for the other four programming languages show the same patterns (§D).

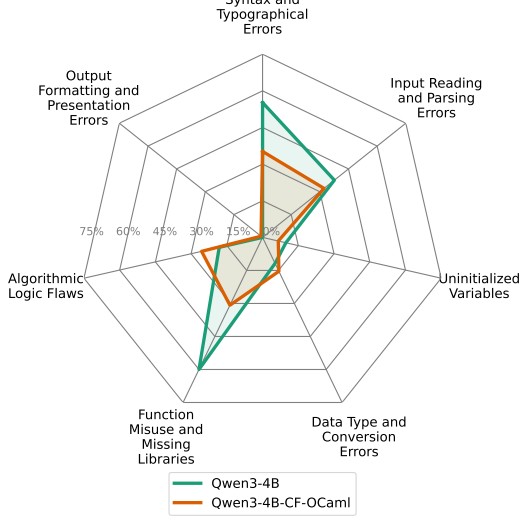

Figure 6: LLM labels of bugs in programs synthesized by Qwen 3 4B and our trained model, Qwen3-4B-CF-OCaml. A radial represents a bug class, and points along the radial show how many programs seem to have that bug. Qwen 3 4B makes more fundamental programming mistakes, such as syntax errors and misusing builtins, showing its limited grasp of OCaml. Our trained model makes far fewer such mistakes. We see a small increase in logic flaws: the trained model makes fewer shallow mistakes, revealing deeper issues.

## 5 CONCLUSION

Arguably, practicioners need LLMs the most for low-resource programming languages, which need highly specialized knowledge. Unfortunately, this is precisely where LLMs are weak due to a lack of sufficient pre-training and post-training data. We propose Agnostics, a language-agnostic post-training pipeline that minimizes the per-language engineering tax by verifying code entirely via externally observable behavior. A short configuration file is enough to adapt the pipeline, including the reinforcement learning setup, to a new programming language.

Empirically, Agnostics consistently improves small open-weight models on five low-resource languages—Lua, Julia, R, OCaml, and Fortran—without requiring language-specific test translators. Training Qwen 3 4B with Ag-Codeforces-X yields large gains on our new Ag-LiveCodeBench-X benchmark and on MultiPL-E, often rivaling or surpassing 16B–70B open-weight baselines. The method scales to larger and different model families: Qwen 3 8B shows similar gains to its smaller sibling, and we also observe improvements on DeepSeek Coder 6.7B, Phi 4 Mini and SmolLM3 3B. Error-type analysis shows our training decreases fundamental programming language mistakes.

A practical advantage of the approach is how little per-language work it requires. After the framework was in place, extending the pipeline to support OCaml and Fortran took us less than an hour each. We expect adaption to be just as straightforward for any pragmatic programming language with a command-line toolchain.

We believe the approach scales to models of arbitrary size, although our experiments are limited by available compute to at most 8B models. For scaling data, the Agnostics reformulation approach also applies to much larger problem sets. For instance, OpenCodeReasoning has ∼600K problems with Python solutions (Ahmad et al., 2025); converting such corpora into language-agnostic I/O tasks would provide rich RL datasets with many target languages with minimal additional engineering.

## REPRODUCIBILITY STATEMENT

We provide references to our data, code, and artifacts at agnostics.abgru.me. We share our datasets and models at huggingface.co/collections/nuprl/agnostics, our Wandb training logs at wandb.ai/nuprl/Agnostics, and our training framework at github.com/nuprl/agnostics-framework.

The existing datasets we used are publicly available and are accompanied by citations. We publicly release all datasets we introduce, allowing free use for research. In the same way, we release all code required for conducting and analyzing our experiments, including the code for dataset preparation, as well as the models we presented and Wandb records from training them. We state the number and range of values tried per (hyper) parameter, and outline how we chose the final values and what they are (§§ 3.3 and C.1). We specify the computing infrastructure (hardware and software) we used for our experiments (§C.5). The released codebases specify the exact versions of all the libraries we used.

## ACKNOWLEDGMENTS

This work is partially supported by the U.S. National Science Foundation (SES-2326174). This material is based upon work supported by the U.S. Department of Energy, Office of Science under Award Number DESC0025613.

This research used resources of the National Energy Research Scientific Computing Center (NERSC), a Department of Energy User Facility using NERSC award ALCC-ERCAP 0038272 (m5083-2024).

This work is supported by Czech Science Foundation Grant No. 23-07580X.

We thank Northeastern Research Computing for support with the Northeastern University Explorer cluster. This work used the Delta cluster at the National Center for Supercomputing Applications (NCSA) through allocation CIS230213 from the Advanced Cyberinfrastructure Coordination Ecosystem: Services & Support (ACCESS) program, which is supported by U.S. National Science Foundation grants 2138259, 2138286, 2138307, 2137603, and 2138296.

This work uses vLLM (Kwon et al., 2023), Transformers (Wolf et al., 2020), and GNU Parallel (Tange, 2023).

*Disclaimer*: This report was prepared as an account of work sponsored by an agency of the United States Government. Neither the United States Government nor any agency thereof, nor any of their employees, makes any warranty, express or implied, or assumes any legal liability or responsibility for the accuracy, completeness, or usefulness of any information, apparatus, product, or process disclosed, or represents that its use would not infringe privately owned rights. Reference herein to any specific commercial product, process, or service by trade name, trademark, manufacturer, or otherwise does not necessarily constitute or imply its endorsement, recommendation, or favoring by the United States Government or any agency thereof. The views and opinions of authors expressed herein do not necessarily state or reflect those of the United States Government or any agency thereof.

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

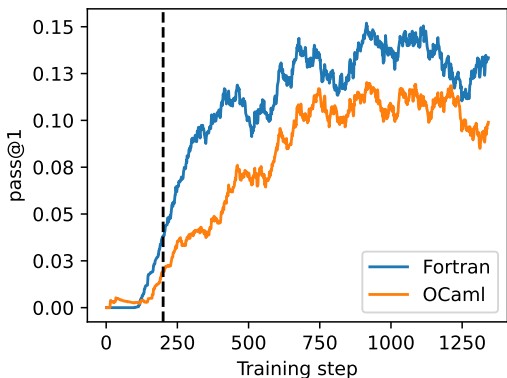

Figure 7: Rewards each step from training Qwen 3 4B on Fortran and OCaml. These are very low-resource languages, and rewards are zero for the first several steps.

## A PREPARING DATASETS FOR AGNOSTICS

**Ag-MBPP-X** is a dataset of Mostly Basic Programming Problems, transformed from the MBPP (Austin et al., 2021) dataset. The source dataset contains problems asking to complete a Python function from on its signature and a docstring, where the docstring specifies what result the function should return; our dataset contains equivalent programs which read/write the same data from/to standard input/output. Out of 974 problems, we are able to translate 776 problems into Ag-MBPP-X (348 problems are in the sanitized subset of MBPP). We analyzed Ag-MBPP-X for data similarity against Ag-LiveCodeBench-X using Decon (2025) and found no data overlap. The analysis was done based on 5-grams with no token sampling (see Decon documentation).

Processing the dataset with Qwen3-32B took less than 1 hour using 2 H100 GPUs. Figures 3a and 3b in the main body of the paper shows a sample problem from the dataset, before and after the reformulation. We used the following prompt.

```
You are a competitive programming expert.
You are given a problem that asks you to implement a function.
Your task is to translate the description of the problem into a form that
    accepts one set of function arguments as inputs and return the
    function return value as output.

Use programming competition style input and outputs -- that is, priorize
    the use of spaces and newlines to separate inputs and outputs over
    using commas and parentheses (or other delimiters). Specifically, for
     2d lists, you should print them as a list of lists, where the outer
    lists elements are separated by newlines and the elements of the
    inner lists are separated by spaces.
For example, a 2d list like [[1, 2], [3, 4]] should be printed as:
1 2
3 4
Do not use any other delimiters.

If there are multiple 2d lists, you should use 2 newlines to separate
    them.
for example, a 2d list like [[1, 2], [3, 4]] and [[5, 6], [7, 8]] should
    be printed as:
1 2
3 4

5 6
7 8

If the problem requires outputing decimal numbers, make sure the output
    format specifies to round all decimal numbers to 4 decimal places. In
```

```
   this case, you should also round all the numbers in the output to 4
   decimal places.

Do not forget to specify the input and output format in the description.

Here is the problem description:
{original mbpp problem description}

Here are the test cases:
{original mbpp test cases}

You should return a json object with the following fields:
- "description": the description of the problem
- "input_format": a string describing the input format
- "output_format": a string describing the output format
- "tests": a list of test cases, each test case is a json object with the
    following fields:
  - "input": a string that represents the input of the test case, in the
    same format as the input format in the description
  - "output": a string that represents the output of the test case, in
    the same format as the output format in the description

Place your response in a single '''json ''' block. Do not include any
    other text in your response.
```

**Ag-Codeforces-X** is a dataset of competitive programming problems, created from Codeforces problems in the `open-r1/codeforces` dataset. The source problems already specified programs by their I/O behavior, hence only very minor changes were needed to build language-universal Agnostics problems out of the fields in the dataset: we only skipped the time and memory restrictions present in the original problems. To be precise, we used data from the `open-r1/codeforces-cots` dataset, `solutions_py_decontaminated` subset, which contains problems decontaminated using 8-gram overlap against multiple benchmarks, in particular LiveCodeBench. We used the auxilliary `checker_interactor` subset to only keep the problems which admit a simple verifier for their solutions, i.e., problems where a single output is correct for each input and where the solution only needs to read data from its input, compute the result, and write it to the standard output. We prepared both a train and a test split. The former contains 5369 problems and the latter contains 105 problems we held out from the source dataset, 5 selected manually and 100 randomly. The manually-selected problems were chosen to be much easier than average, to make it easier to detect if a model can solve any problems at all in a given programming language. In short, the 5 problems are: "output a number in binary notation", "remove all digits from a string", "check if the parentheses are balanced", "parse two integers and add them", and the following longer problem: "Petr stands in line of n people, but he doesn't know exactly which position he occupies. He can say that there are no less than a people standing in front of him and no more than b people standing behind him. Find the number of different positions Petr can occupy."

The train split features randomized prompts, which we found help with generalizing the results of training on the dataset to other benchmarks. The prompts were randomly split into a number of types. 30% of the prompts use standard Markdown headings to start different sections of the prompt, 35% use bold text instead, and the remaining 35% simply concatenate the prompt sections together. Most of the prompts follow the source dataset and feature an I/O sample in the prompt, with other samples withheld as private. Half of the prompts of the final type do not feature any I/O sample.

**Ag-LiveCodeBench-X** is also a dataset of competitive programming problems, created from a subset of the LiveCodeBench dataset (Jain et al., 2024a). LiveCodeBench 5.0 has 880 problems, of which 381 have Python starter code and test cases. The remaining 499 problems do not use starter code and instead use standard I/O to specify and test solutions. Hence we used these problems to transform LiveCodeBench into an Agnostics dataset. Ag-LiveCodeBench-X only has a test split, like its source dataset.

# B AGNOSTICS CONFIGURATIONS

In this section, we list the configurations that we use for our target languages. The configuration files use YAML. The prompts for OCaml and Fortran have instructions generated by OpenAI o3.

The Lua configuration:

```
prompt: Use Lua 5.1, targeting LuaJIT.
install: apt-get install -y luajit
filename: snippet.lua
execute: luajit snippet.lua
```

The Julia configuration:

```
prompt: Use Julia 1.11.
container:
  base-image: "julia:1.11.3"
  type: debian
filename: snippet.jl
execute: julia snippet.jl
```

The R configuration (unmodified, unlike Figure 4):

```
install: apt-get install -y r-cran-tidyverse
filename: snippet.R
execute: Rscript snippet.R
prompt: |
  Use R version 4. Use `readLines(con = file("stdin"))` to read input
    from stdin. Optionally, use the `n` argument to read the first `n`
    lines. For example:
  ```r
  input <- readLines(con = file("stdin"), n = 1)
  n <- as.integer(input)
  cat(n) # print the first line of input
  ```
  Also, use `cat` to print output to stdout. For example:
  ```r
  cat(n)
  ```
  Please do not use `print` to print output.
```

The OCaml configuration:

```
prompt: |
  Use OCaml 5.

  Numbers:   + - * / mod   vs.   +. -. *. /. **    (add dots!)
  Casts:     float_of_int   int_of_float   int_of_string
  Mutation:  refs (:= !) or pass new values recursively
  Strings:   split_on_char, String.get => char, use Printf "%c"
  Lists:     avoid List.nth; prefer pattern-match / folds / arrays
container:
  base-image: "docker.io/ocaml/opam:ubuntu-22.04-ocaml-5.0"
  type: debian
install:
  container-instructions: |
    RUN opam install base stdio utop
    ENV OPAM_SWITCH_PREFIX='/home/opam/.opam/5.0'
    ENV CAML_LD_LIBRARY_PATH='/home/opam/.opam/5.0/lib/stublibs:/home/
    opam/.opam/5.0/lib/ocaml/stublibs:/home/opam/.opam/5.0/lib/ocaml'
    ENV OCAML_TOPLEVEL_PATH='/home/opam/.opam/5.0/lib/toplevel'
    ENV MANPATH=':/home/opam/.opam/5.0/man'
```

```
    ENV PATH='/home/opam/.opam/5.0/bin:/usr/local/sbin:/usr/local/bin:/
    usr/sbin:/usr/bin:/sbin:/bin'
filename: snippet.ml
execute: utop -require base -require stdio snippet.ml
```

The Fortran configuration:

```
prompt: |
  Use Fortran 90. Some tips:

  Always begin each scope with implicit none, pick explicit kinds via
    selected_*_kind, and declare proper lengths-character(len=*) is legal
     only for dummy arguments, not locals.  Strings are blank-padded:
    call len_trim before iterating, and store dynamic text in deferred-
    length allocatables (character(len=:), allocatable :: s).  List-
    directed read(*,*) arr does not auto-size arrays; read a count first,
     then allocate and read, or tokenize a line manually.  When
    translating 0-based formulas (heaps, bit positions) remember Fortran
    arrays default to 1-based; if you want 0-based, declare lower bounds.
     Use real literals (2.0d0, 1.0_rk) to avoid silent integer division,
     and guard against overflow when exponentiating integers.  For
    frequency tables, allocate an array or use findloc; Fortran lacks
    native dicts/sets, so you must implement search yourself.  Prefer
    array intrinsics (sum, count, pack) over hand-rolled loops, and keep
    helper procedures inside a contains section or module so interfaces
    are explicit.  return inside the main program is non-idiomatic; use
    structured blocks or stop.  Never print interactive prompts in batch
    solutions; just read, compute, and write.
install: apt-get install -y gfortran
filename: snippet.f90
compile: gfortran -o snippet.out snippet.f90
execute: ./snippet.out
```

## C  TRAINING AND RESULTS

### C.1  CHOOSING HYPERPARAMETERS

Before picking the hyperparameters described in §3.3, we investigated other values by training the Qwen 3 4B model on a previous version of Ag-Codeforces-X. We trained two models for each of Lua, Julia and R, using a linear learning rate schedule with the same learning rate. We decided against it since some of the runs degraded the model's capabilities, unlike any of the runs we did with a cosine decay schedule.

We compared between GRPO group sizes of 16, 32 and 64 by training the same model on Lua. In some runs with group size 16, we saw the model improved significantly less than at higher group sizes. We ran an experiment to compared different temperature and group size settings (Table 6).

The models trained at group size 16 had slightly lower scores compared to other models, while the ones trained at group size 64 displayed scores comparable to other models. However, they took significantly longer to train. Two group size 64 models took $\sim 20.5$h to train on average (the third one was trained on a different machine). In comparison, the group size 32 models trained at the same time on the same machine took $\sim 12$h on average. The models trained at temperatures other than the 0.7 recommended by the Qwen team performed similarly to the other models.

As we found no significant difference between the temperature settings and between group sizes 32 and 64, we chose the smaller group size due to limited resources, and used the recommended temperature settings.

### C.2  TRAINING DYNAMICS

In this section we discuss the measurements we took while training the models. Figure 8 shows the GRPO group batch pass@1 while training the Qwen3-4B-CF-X models. The scores of all the

Table 6: Hyperparameter sweep, pass@1 score.

| Model | Group size | Temperature | Ag-LiveCodeBench-X |
|---|---|---|---|
| Qwen3-4B-CF-Lua | 32 | 0.7 | 23.00 |
| normal-r1 | 32 | 0.7 | 19.87 |
| normal-r2 | 32 | 0.7 | 21.58 |
| size16-r1 | 16 | 0.7 | 19.80 |
| size16-r2 | 16 | 0.7 | 19.40 |
| size16-r3 | 16 | 0.7 | 19.65 |
| size16-r4 | 16 | 0.7 | 20.61 |
| size64-r1 | 64 | 0.7 | 21.71 |
| size64-r2 | 64 | 0.7 | 21.21 |
| size64-r3 | 64 | 0.7 | 20.87 |
| temp0p2-r1 | 32 | 0.2 | 19.91 |
| temp0p2-r2 | 32 | 0.2 | 20.39 |
| temp0p2-r3 | 32 | 0.2 | 21.98 |
| temp1-r1 | 32 | 1.0 | 21.22 |
| temp1-r2 | 32 | 1.0 | 21.48 |
| temp1-r3 | 32 | 1.0 | 20.30 |

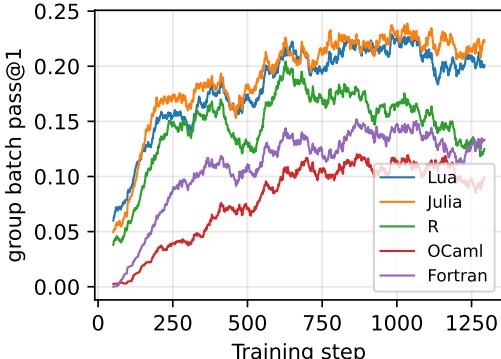

Figure 8: Training Qwen3-4B-CF-X, GRPO group batch pass@1.

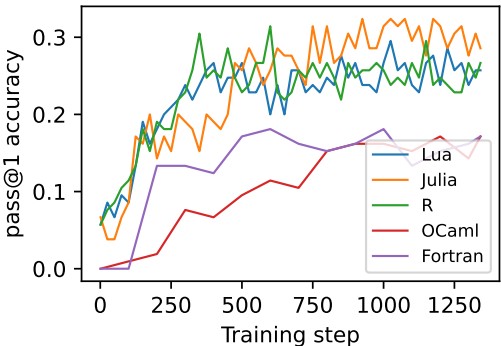

Figure 9: Training Qwen3-4B-CF-X, test split pass@1.

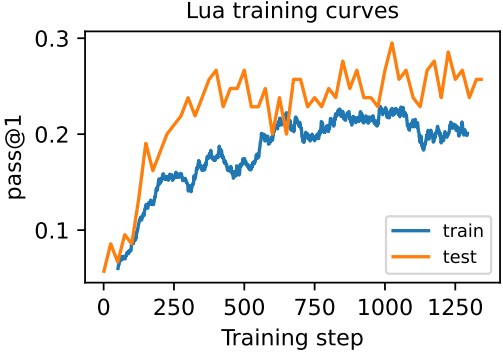

Figure 10: Training Qwen3-4B-CF-Lua, GRPO group batch and test split pass@1.

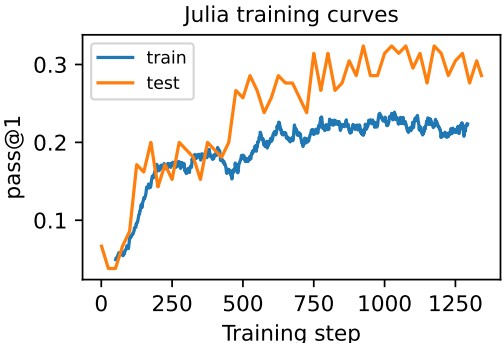

Figure 11: Training Qwen3-4B-CF-Julia, GRPO group batch and test split pass@1.

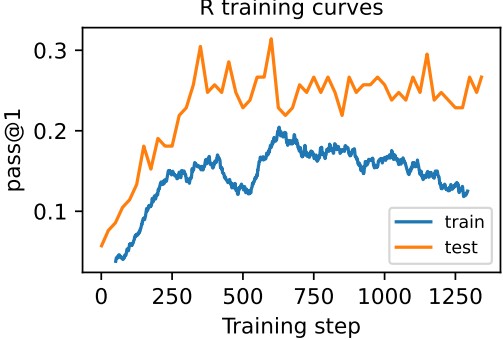

Figure 12: Training Qwen3-4B-CF-R, GRPO group batch and test split pass@1.

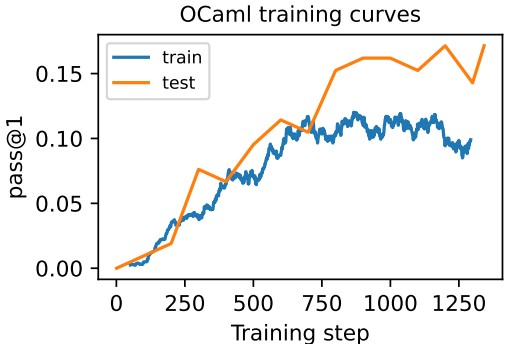

Figure 13: Training Qwen3-4B-CF-OCaml, GRPO group batch and test split pass@1.

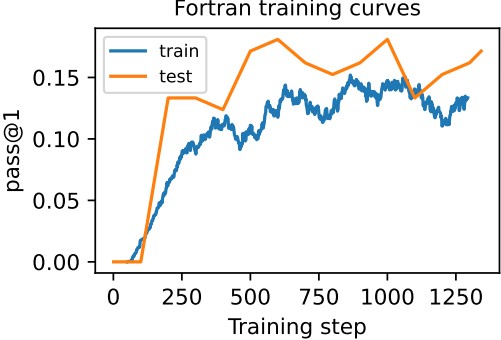

Figure 14: Training Qwen3-4B-CF-Fortran, GRPO group batch and test split pass@1.

Table 7: Partial reward experiment, pass@1 rates.

| Model | Ag-LiveCodeBench-X | Ag-Codeforces-X (test split) |
|---|---|---|
| Qwen3-4B-CF-Lua | 23.00 | 24.76 |
| partial-r1 | 18.57 | 13.33 |
| partial-r2 | 20.16 | 15.62 |

models are broadly correlated with one another, which may at least in part be due to training them on the same permutation of the training data. Figures 10 to 14 compare the GRPO group pass@1 scores with the pass@1 scores on the test split. We see that the scores on the test split is broadly correlated with the train split rewards. In most cases, we see that the train scores keep increasing until the end of the epoch, together with the test split pass@1 scores, indicating that the model keeps improving until the end of the dataset.

### C.3 REWARD FUNCTION

We investigated the results of partial rewards. We trained Qwen 3 4B—the base model of Qwen3-4B-CF-X—on Ag-Codeforces-Lua, giving it a partial reward of $0.2$ if it generated code which failed one of the tests by producing wrong output but otherwise terminated without an error. The full reward for a snippet passing all tests was still 1. Table 7 shows that the trained models score below Qwen3-4B-CF-Fortran both on Ag-LiveCodeBench-Lua and on the test split of Ag-Codeforces-Lua (counting only the full-credit reward). The latter scores are particularly far lower, clearly showing that the models learned to abuse the partial-credit reward.

During training, we saw the models focus on the partial reward. The average result from the partial reward component was clearly increasing more quickly than the result from the full reward component. In the training generations we inspected, the models also often claimed to generate a "draft" answer and produced a program which ignored the problem in the prompt, for instance by only printing a hard-coded string such as "0".

### C.4 CROSS-PROGRAMMING-LANGUAGE NEGATIVE TRANSFER

To demonstrate that Agnostics training does not lower performance on different programming languages, we evaluated the models we trained on variants of Ag-LiveCodeBench-X (Table 8).

### C.5 HARDWARE AND SOFTWARE USED

We used three machines while working on this paper: B, R1 and R2. R2 was only used to generate completions of trained models for evaluation, while B and R1 were used to train models. Upon publication of the paper, we will publicly release Wandb records of our training runs, which include the duration and the machine used.

B has 2 Intel Xeon Gold 6342 CPUs @ 2.80GHz, 1008 GB of RAM, 4 NVIDIA H100 80GB, and uses Ubuntu 22.04.5 LTS.

R1 has 2 AMD EPYC 9454 48-Core CPUs, 8 NVIDIA H100 80GB (with NVLink connections), 2268 GB of RAM, and uses Ubuntu 22.04.5 LTS.

R2 has 2 Intel(R) Xeon(R) Gold 6326 CPU @ 2.90GHz, 10 NVIDIA RTX A600, 504 GB of RAM, and uses Ubuntu 22.04.5 LTS.

When developing the Agnostics framework, we used the following major Python libraries: `ray v2.46.0, torch v2.6.0, transformers v4.54.1, vllm v0.8.5.post1, datasets v3.4.1,wandb 0.19.11`.

Table 8: Cross-PL evaluation, pass@1 rates.

| Model | Ag-LiveCodeBench-X | | | |
|---|---|---|---|---|
| X= | Python | Lua | Julia | R |
| Qwen 3 4B | 34.34 | 11.00 | 10.00 | 10.00 |
| Qwen3-4B-CF-Lua | 32.96 | 23.00 | 6.55 | 3.00 |
| Qwen3-4B-CF-Julia | 35.10 | 8.43 | 22.00 | 3.90 |
| Qwen3-4B-CF-R | 31.58 | 9.08 | 7.92 | 15.00 |
| Qwen 3 8B | 33.51 | 11.00 | 9.00 | 9.00 |
| Qwen3-8B-CF-Lua | 34.29 | 25.00 | 7.44 | 5.96 |
| Qwen3-8B-CF-Julia | 33.84 | 8.26 | 25.00 | 7.03 |
| Qwen3-8B-CF-R | 34.49 | 9.90 | 7.34 | 19.00 |
| DSC 6.7B Ins | 16.86 | 7.89 | 4.79 | 7.97 |
| DSC-6.7B-Ins-CF-Lua | 17.63 | 8.93 | 8.60 | 8.08 |
| DSC-6.7B-Ins-CF-Julia | 17.60 | 9.54 | 9.13 | 8.56 |
| DSC-6.7B-Ins-CF-R | 17.62 | 8.38 | 5.75 | 9.83 |
| Phi4 mini ins | 19.87 | 7.95 | 8.10 | 4.69 |
| Phi4-mini-ins-CF-Lua | 22.16 | 11.80 | 8.02 | 4.56 |
| Phi4-mini-ins-CF-Julia | 21.15 | 9.10 | 7.69 | 4.22 |
| Phi4-mini-ins-CF-R | 19.82 | 8.02 | 8.54 | 11.54 |
| SmolLM3 3B | 20.91 | 1.02 | 2.85 | 2.07 |
| SmolLM3-3B-CF-Lua | 21.81 | 7.46 | 2.93 | 2.17 |
| SmolLM3-3B-CF-Julia | 21.58 | 1.53 | 7.83 | 2.39 |
| SmolLM3-3B-CF-R | 21.63 | 1.30 | 3.30 | 5.90 |

## D  BUG TAXONOMY

### D.1  PROMPT FOR GENERATING TAXONOMY

We used the following instructions to generate the bug taxonomy, followed by a list of faulty R programs.

> **Input:** The attached file contains multiple failed R programs (Version 4) with their:
>
> - Source code
> - Expected output
> - Actual standard output
> - Error messages (where applicable)
>
> **Objective:** Analyze these program failures systematically to create a comprehensive taxonomy of 10-12 bug themes that categorize the underlying causes of failure.
>
> **Instructions:**
>
> 1. **Initial Analysis**
>    - Read through ALL program examples carefully
>    - For each failure, identify the root cause (not just the symptom)
>    - Note any patterns or commonalities across failures
> 2. **Taxonomy Development**
>    - Create 10-12 distinct bug themes that collectively cover all observed failures
>    - Each theme should represent a fundamental type of programming error or misconception
>    - Themes should be mutually exclusive when possible, but comprehensive in coverage
>    - Order themes from most to least frequent (or by logical grouping)
> 3. **For Each Bug Theme, Provide:**

- Theme Name: A concise, descriptive title
- Description: 2-3 sentences explaining the nature of this bug type
- Common Symptoms: How these bugs typically manifest (error messages, incorrect output, etc.)
- Root Causes: The underlying programming mistakes or misconceptions
- Examples: Reference 2-3 specific programs from the file that exhibit this theme
- Prevention Tips: Brief advice on how to avoid this type of bug

4. **Constraints:**
   - Focus on R-specific issues as well as general programming errors
   - Base your taxonomy ONLY on the provided examples
   - You may search online ONLY to understand specific R error messages or function behavior, not for existing bug taxonomies
   - Ensure every failed program in the file can be classified under at least one theme

5. **Deliverable Format:** Present your taxonomy as a numbered list with clear formatting and comprehensive coverage of all observed failure patterns. Supply a short explanation for each theme in your taxonomy.

The prompt produced a taxonomy of 11 bug categories. We edited these categories and selected 7 categories relevant to us, shown in §D.2.

## D.2    Bug Taxonomy Used For Analysis

The following categories represent the prevalent themes of programming errors we use in our analysis of bugs in model-generated code. They cover the full spectrum of parse, runtime, and logical failures typically encountered in programming. The themes are not mutually exclusive; we allow a program to have more than one themes.

1. **Syntax and Typographical Errors**: Missing commas, mismatched parentheses, or other typos that cause compile-time parse errors.

2. **Input Reading and Parsing Errors**: Mis-reading or mis-parsing input, leading to empty or malformed variables and subsequent failures.

3. **Uninitialized Variables**: References to variables never defined, causing undefined behavior or runtime faults.

4. **Data Type and Conversion Errors**: Incorrect casting or type misuse that triggers type errors, warnings, or incorrect results.

5. **Function Misuse and Missing Libraries**: Invocations of non-existent or mis-parameterized functions, or missing imports/libraries, causing errors.

6. **Algorithmic Logic Flaws**: Programs that compile and run but produce wrong answers due to faulty logic or conditions.

7. **Output Formatting and Presentation Errors**: Correct computational results, but incorrect due to formatting issues (e.g. missing newlines/spaces or output spec violations).

## D.3    Radar Charts For All Programming Languages

Figures 15, 16, 17, 18, 19 show the error theme charts for all the programming languages we trained a model on. Figure 18 is the same as Figure 6 from the main body; we repeated it here for convenience.

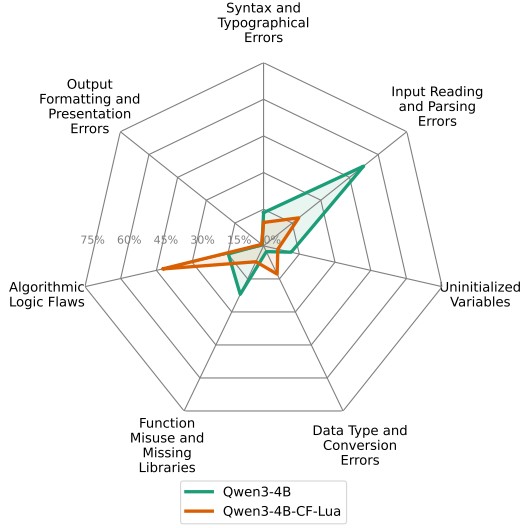

Figure 15: Radar chart of Lua error themes for Qwen3-4B and Qwen3-4B-CF-Lua.

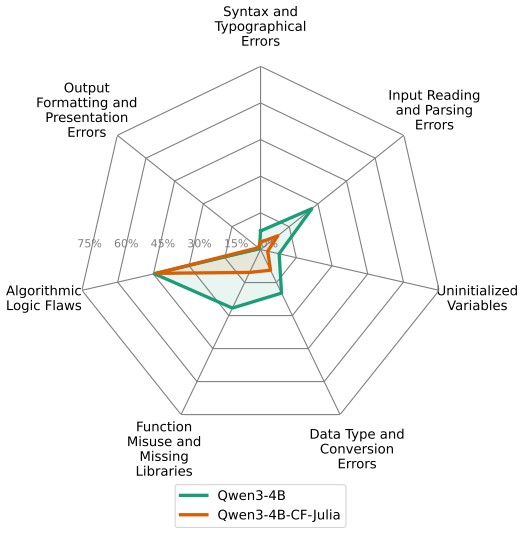

Figure 16: Radar chart of Julia error themes for Qwen3-4B and Qwen3-4B-CF-Julia.

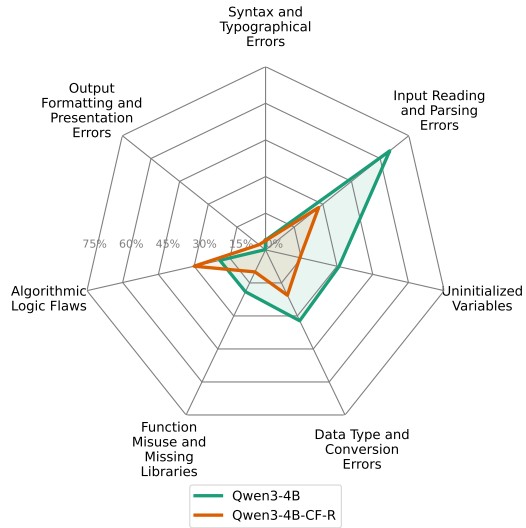

Figure 17: Radar chart of R error themes for Qwen3-4B and Qwen3-4B-CF-R.

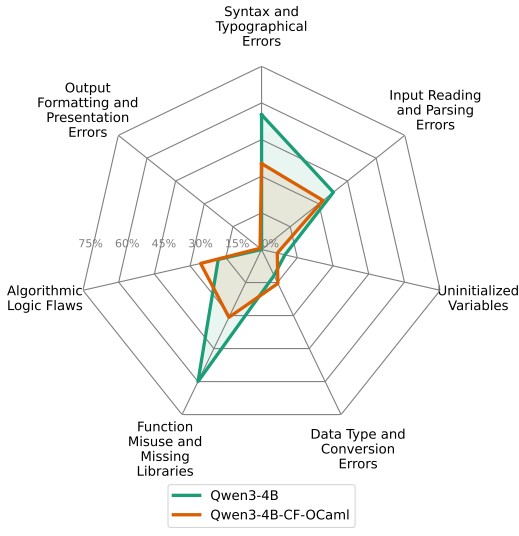

Figure 18: Radar chart of OCaml error themes for Qwen3-4B and Qwen3-4B-CF-OCaml.

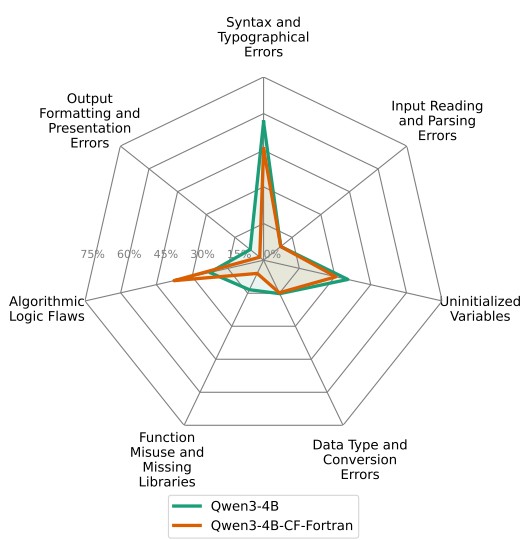

Figure 19: Radar chart of Fortran error themes for Qwen3-4B and Qwen3-4B-CF-Fortran.

