# OpenReview forum: "Agnostics: Learning to Synthesize Code in Any Programming Language with a Universal Reinforcement Learning Environment"
_ICLR.cc/2026/Conference — ICLR 2026 Poster_

### Official Review · Reviewer_pRJF · 2025-10-28

**Soundness:** 3
**Presentation:** 3
**Contribution:** 3
**Rating:** 8
**Confidence:** 4

**Summary:**

The paper proposed a training pipeline that can be used in any programming language with only a small configuration file by describing the input and output specification in pure text format.

* The proposed method does not require a language-specific verifier to verify and provide the rewards but rather evaluate the model on the observable behavior, in this paper, the text output.
* It provides several datasets that have only I/O description and a suite of tests with only input and output. in text format.
* It provides a high optimized GRPO training framework with code execution sandbox.
* It provides a SOTA open-weight 4B model on low resource language under 16B size.

**Strengths:**

* The training pipeline address the task from a different direction where only evaluating on the input and output, so it can be applied to any language without a language specific verifier. And specifically, reward on the format is offloaded into a prompt prefix instead to instruct the model to follow the format.
* In low resource languages, the claimed method shows good improvement on small models and even beating bigger models trained specifically on the code task.
* The paper is well written, it constructs a clear flow from problem to method to the final evaluation. The experiments, setup and training hyperparameters are described clearly.
* The paper does a thorough analysis on the approach, including the impact on the different model architecture, model size, training data and fine grained analysis on models’ error types.

**Weaknesses:**

* Given the importance that a good prefix helps avoid the **most common** errors and if the model barely knows a programming language per line 199, and you select several faulty generation for generating the verbatim, I image this has to be representative so that the model can generate a verbatim that can avoid the **most common** errors, so I think it’s necessary to include a description on how to select the faulty generations.
* Line 76 claims the paper provides a small and highly-optimized training framework, but there is no analysis/experiments/comparison to demonstrate.

**Questions:**

* Line 240, miss the definition of the $r_{i,t}(\theta)$
* Line 255, please explain OCI
* Line 335. how is the score defined?
* Line 420, can you clarify the $96.
* Line 424, what’s the epochs used in table 2 for Qwen3-4B-CF-Fortran.
* Line 916, what's the 7 manually selected problems and how is it selected?
* Do you have a analysis on the data similarity between CF-X and LCB-X, same for the MBPP-X and LCB-X. This is important to exclude the factor from data similarity in the score improvement.

---

> ### Author Response · Authors · 2025-11-21
>
> We thank you for your detailed and thorough review! We’re happy to see you appreciate our work. We answer your concerns and questions below.
> ### Explaining how to prepare a good prefix.
>
> > Given the importance that a good prefix helps avoid the most common errors and if the model barely knows a programming language per line 199, and you select several faulty generation for generating the verbatim, I image this has to be representative so that the model can generate a verbatim that can avoid the most common errors, so I think it’s necessary to include a description on how to select the faulty generations.
>
> We apologize for the unfortunate unclear phrasing, we meant that the selected errors only need to be common enough. We clarified the text in the paper. To prepare the prefix, we picked the first few code snippets which did not compile, and asked a capable LLM to select the most common errors in those particular snippets. It was enough to put the baseline barely above 0% for reinforcement learning to start working. For the programming languages we investigated, we saw no need to carefully analyze which errors are the most common across all possible generations.
> ### Training framework claims.
>
> > Line 76 claims the paper provides a small and highly-optimized training framework, but there is no analysis/experiments/comparison to demonstrate.
>
> We adjusted the text to instead call our framework "small and carefully designed". Our framework is more focused compared to codebases like HuggingFace’s `trl`. Our trainer and container orchestration modules are only 1023 and 830 LoC (excluding blank lines and comments).
> ### Definition of the objective.
>
> > Line 240, miss the definition of the $r_{i,t}(\theta)$
>
> We apologize, we fixed the definition. Due to Latex typos, the original submission by mistake included an objective definition with missing parts.
> ### OCI
>
> > Line 255, please explain OCI
>
> We cited <https://opencontainers.org/>.
>
> ### Score definition.
>
> > Line 335. how is the score defined?
>
> We added a reference to Section 4, at the start of which we explain that “we measure pass@1 accuracy with reasoning disabled, 20 samples per prompt at temperature 0.2”.
> ### Experiment cost description.
>
> > Line 420, can you clarify the $96.
>
> We explained in the paper that we meant to show that the training data was not generated using significantly more compute compared to Agnostics training; the cost is used as a rough measure.
> ### Number of training epochs.
>
> > Line 424, what’s the epochs used in table 2 for Qwen3-4B-CF-Fortran.
>
> Unless otherwise specified, we trained the models on 1 epoch. We clarified this at the start of Sec. 4.
> For clarity, we also briefly mentioned Ag-MBPP-X in Sec. 4.1.
> ### Manually selected test split problems.
>
> > Line 916, what's the 7 manually selected problems and how is it selected?
>
> They were chosen to be much easier than other problems in the dataset, to facilitate testing if the model can solve any problems at all in a given programming language. Unfortunately there was a typo, we only manually chose 5 problems. We fixed the typo and we added the following description to the appendix.
>
> In short, the problems are: “output a number in binary notation”, “remove all digits from a string”, “check if the parentheses are balanced”, “parse two integers and add them”, and the following longer problem: “Petr stands in line of n people, but he doesn't know exactly which position he occupies. He can say that there are no less than a people standing in front of him and no more than b people standing behind him. Find the number of different positions Petr can occupy.”
> ### Data similarity analysis.
>
> > Do you have a analysis on the data similarity between CF-X and LCB-X, same for the MBPP-X and LCB-X. This is important to exclude the factor from data similarity in the score improvement.
>
> Appendix B explains how the dataset used to make Ag-Codeforces-X was decontaminated to avoid overlap with LiveCodeBench: “[the Codeforces dataset] contains problems decontaminated using 8-gram overlap against multiple benchmarks, in particular LiveCodeBench”.
>
> Regarding Ag-MBPP-X, we extended appendix “Preparing Datasets for Agnostics” with the results of an ngram-based contamination analysis against Ag-LiveCodeBench-X using [decon](https://github.com/allenai/decon) and found zero overlap. The exact analysis is documented [here](https://github.com/allenai/decon/blob/main/doc/simple.md); we used 5-grams with no token sampling.

---

### Official Review · Reviewer_oRWA · 2025-10-31

**Soundness:** 3
**Presentation:** 3
**Contribution:** 3
**Rating:** 4
**Confidence:** 4

**Summary:**

This paper introduce Agnostics, a programming language-agnostic LLM post-training pipeline. The main idea is to rewrite the coding questions into a question that are programming agnostic and only works on standard input/output. Then, these questions can be further rewritten into programming language specific questions. Experiment results show that Agnostic can significantly improve the model performance in low resource programming languages.

**Strengths:**

1. The proposed idea is quite simple and elegant. With the proposed approach, we can theoretically turn coding problems in any programming languages into other languages.
2. It achieves promising results on single language training, showing that models trained with such data achieves much better performance in low resource languages. What's more interesting here is that the improvement generalizes beyond standard I/O problems.

**Weaknesses:**

1. The number of programming languages tested are limited (5 of them). Moreover, it is only trained on data from a single programming language in each experiment. This is not a usual setup. Ideally, we would like to mix data from all programming languages together (high and low resources), and see how the performance of high/low resource languages change.
2. Unfortunately the model scale is also limited (up to 8B). So we cannot verify if this approach can scale well with model size.

**Questions:**

My main concern is that the training is not performed on data from all languages mixing together. Can you show such results? We would like to verify that there are no negative transfer across programming languages and the approach is indeed pushing the frontier.

---

> ### Author Response · Authors · 2025-11-21
>
> Thank you for your review, we appreciate that found our approach elegant! We address your main concern below.
>
> ### Lack of negative transfer across programming languages.
>
> To show there is no negative transfer across languages, we evaluated the Qwen3-4B-CF-X and Qwen3-8B-CF-X models on Ag-LiveCodeBench-Python. They all scored similarly to the base model, confirming our observations while working on this project. We can repeat this experiment for more models and programming languages. (EDIT: see a more comprehensive experiment in the following comment.)
>
> | Model               | PL | pass@1  |
> |---------------------|----|--------|
> | Qwen3 4B            | py | 0.3434 |
> | Qwen3 8B            | py | 0.3351 |
> | Qwen3-4B-CF-Fortran | py | 0.3364 |
> | Qwen3-4B-CF-Julia   | py | 0.3510 |
> | Qwen3-4B-CF-Lua     | py | 0.3296 |
> | Qwen3-4B-CF-Ocaml   | py | 0.3477 |
> | Qwen3-4B-CF-R       | py | 0.3158 |
> | Qwen3-8B-CF-Julia   | py | 0.3384 |
> | Qwen3-8B-CF-Lua     | py | 0.3429 |
> | Qwen3-8B-CF-R       | py | 0.3449 |

---

> > ### Author Response · Authors · 2025-12-03
> >
> > ### Lack of negative transfer across programming languages.
> > We extended the experiment,
> > added its results to the "Training and Results" appendix,
> > and referenced them in the "Results" section (4.3).
> >
> > Evaluating all but two models we trained across Python, Lua, Julia and R,
> > we see no evidence of negative cross-programming-language transfer.
> >
> > | Model                     | Python | Lua    | Julia  | R      |
> > |---------------------------|--------|--------|--------|--------|
> > | Qwen3 4B                  | 0.3434 | 0.1100 | 0.1000 | 0.1000 |
> > | Qwen3-4B-CF-Lua           | 0.3296 | 0.2300 | 0.0655 | 0.0300 |
> > | Qwen3-4B-CF-Julia         | 0.3510 | 0.0843 | 0.2200 | 0.0390 |
> > | Qwen3-4B-CF-R             | 0.3158 | 0.0908 | 0.0792 | 0.1500 |
> > | Qwen3 8B                  | 0.3351 | 0.1100 | 0.0900 | 0.0900 |
> > | Qwen3-8B-CF-Lua           | 0.3429 | 0.2500 | 0.0744 | 0.0596 |
> > | Qwen3-8B-CF-Julia         | 0.3384 | 0.0826 | 0.2500 | 0.0703 |
> > | Qwen3-8B-CF-R             | 0.3449 | 0.0990 | 0.0734 | 0.1900 |
> > | DSCoder 6.7B Ins          | 0.1686 | 0.0789 | 0.0479 | 0.0677 |
> > | DSCoder-6.7B-Ins-CF-Lua   | 0.1763 | 0.0893 | 0.0860 | 0.0733 |
> > | DSCoder-6.7B-Ins-CF-Julia | 0.1760 | 0.0954 | 0.0913 | 0.0779 |
> > | DSCoder-6.7B-Ins-CF-R     | 0.1762 | 0.0838 | 0.0575 | 0.0858 |
> > | Phi4 mini ins             | 0.1987 | 0.0795 | 0.0810 | 0.0052 |
> > | Phi4-mini-ins-CF-Lua      | 0.2216 | 0.1180 | 0.0802 | 0.0017 |
> > | Phi4-mini-ins-CF-Julia    | 0.2115 | 0.0910 | 0.0769 | 0.0038 |
> > | Phi4-mini-ins-CF-R        | 0.1982 | 0.0802 | 0.0854 | 0.0065 |
> > | SmolLM3 3B                | 0.2091 | 0.0102 | 0.0285 | 0.0000 |
> > | SmolLM3-3B-CF-Lua         | 0.2181 | 0.0746 | 0.0293 | 0.0000 |
> > | SmolLM3-3B-CF-Julia       | 0.2158 | 0.0153 | 0.0783 | 0.0000 |
> > | SmolLM3-3B-CF-R           | 0.2163 | 0.0130 | 0.0330 | 0.0000 |

---

### Official Review · Reviewer_ZVFQ · 2025-11-01

**Soundness:** 3
**Presentation:** 3
**Contribution:** 2
**Rating:** 4
**Confidence:** 3

**Summary:**

The paper introduces Agnostics, a language-agnostic RL pipeline that reformulates coding tasks into standard I/O problems and trains code LLMs using verifiable rewards in a containerized sandbox. With small per-language configs, the same verifier supports many languages. Applied to Lua, Julia, R, OCaml, and Fortran, Agnostics boosts small open models to match/exceed larger baselines.

**Strengths:**

•	Consistent gains across five low-resource languages with small models.

•	New multi-language benchmark and dataset.

**Weaknesses:**

1.	While the contribution is clearly useful and highly practical, my main concern with the paper is that the contribution is more of an engineering improvement rather than a fundamental scientific contribution grounded in theoretical or algorithmic insights.

2.	Appendix A argues that rejection sampling is prohibitively expensive on hard tasks. A direct empirical comparison would clarify efficiency and quality trade-offs

3.	The paper would benefit from a more detailed ablation study: (i) reintroduce a KL term and show effects on stability/generalization; (ii) quantify partial credit reward pitfalls; (iii) temperature/group size sweeps beyond the brief D.1 notes. This would strengthen the methodological contribution beyond the system build.

**Questions:**

•	Can you elaborate more on what is the contribution of the paper? Can you clarify the fundamental, theoretical or algorithmic level contributions?

•	Did you perform any ablation studies related to KL term and partial-credit reward?

---

> ### Author Response · Authors · 2025-11-21
>
> Thank you for your careful review. We address your questions below.
>
> ### Elaborating on our contributions.
>
> > Can you elaborate more on what is the contribution of the paper? Can you clarify the fundamental, theoretical or algorithmic level contributions?
>
> We implement and study a novel technique for preparing reinforcement learning environments which allows training a model to code in diverse low-resource programming languages much more easily than previously known approaches. We train very capable models for code generation, performing the best among ${\leq}15B$ parameter open-weights models. We release our framework and training datasets together with Ag-LiveCodeBench-X, a new multi-language benchmark which is both harder than comparable pre-existing benchmarks and easier to extend with other programming languages.
>
> From the fundamental research perspective, our work facilitates and influences further research related to low-resource programming languages, e.g., it arguably means it is more meaningful to investigate effects such as [crosslingual generalization](http://arxiv.org/abs/2211.01786).
>
> ### Empirical comparison with rejection sampling.
>
> > Appendix A argues that rejection sampling is prohibitively expensive on hard tasks. A direct empirical comparison would clarify efficiency and quality trade-offs
>
> We updated the paper with a new experiment that directly compares our approach to rejection sampling. Using Qwen 3 4B, we sampled responses to problems in the train split of Ag-Codeforces-Fortran with the same settings we used to train Qwen3-4B-CF-Fortran. In total we took 171808 samples.
>
> The base model succeeded 0.09% of the time, generating 158 test-passing programs. Fine-tuning on these 158 programs has no impact on model performance. In contrast, during Agnostics training Qwen3-4B-CF-Fortran generated 11400 test-passing programs overall.
>
> ### KL term and partial-credit reward ablations.
>
> > Did you perform any ablation studies related to KL term and partial-credit reward?
>
> Regarding the KL term, a number of works (e.g. [Understanding R1-Zero-like Training](https://arxiv.org/abs/2503.20783), [Open-Reasoner-Zero](https://arxiv.org/abs/2503.24290), [DAPO](https://arxiv.org/abs/2503.14476)), found that removing the term is beneficial or at least neutral to training a model. Given that computing the KL term is computationally expensive as it involves an additional model, we did not perform ablation studies involving the term.
>
> Regarding partial-credit reward, we extended the appendix “Training and Results” with the results of the following experiment (new D.3). We trained two models based on Qwen 3 4B using Ag-Codeforces-Lua, giving them a partial reward of 0.2 for Lua snippets which failed a test by producing wrong output but otherwise terminated with no error. The full reward for a snippet passing all tests was still 1. On Ag-LiveCodeBench-Lua, the trained models scored 18.57% and 20.16%, lower than ~23% reached by Qwen3-4B-CF-Fortran.
>
> The models prioritized the partial reward during training (EDIT: as shown in a table in the following comment).
> The average result from the partial reward component was clearly increasing more quickly than the result from the full reward component.
> The models also often claimed to generate a “draft” answer and produced a program ignoring the prompt and only printing a fixed string.
>
> ### Temperature and group size sweeps.
>
> > The paper would benefit from a more detailed ablation study: (i) (...); (ii) (...); (iii) temperature/group size sweeps beyond the brief D.1 notes.
>
> We extended D.1 with a sweep of group size and temperature settings, training Qwen 3 4B on Ag-Codeforces-Lua and evaluating the final models on Ag-LiveCodeBench-Lua.
> The appendix includes a table which does not fit here due to the response limits.
> (EDIT: for convenience we copied the table in a following comment.)
> We trained 2 models at group size 32 and temperature 0.7 (same as Qwen3-4B-CF-Lua),
> 4 models at group size 16,
> 3 models at group size 64,
> 3 models at temperature 0.2
> and 3 models at temperature 1.0,
> 15 models in total.
>
> The group size 16 models had slightly lower scores compared to other models. They did not display dips in reward during training, unlike similar models we trained before.
>
> The group size 64 models displayed scores comparable to other models. They took much longer to train, further motivating using group size 32 for the main experiments. Two group size 64 models took ~20.5h to train on average (the third one used another machine). In comparison, the group size 32 models trained concurrently on the same machine took ~12h on average.
>
> The models trained at different temperature settings scored comparably to models trained at the same settings as Qwen3-4B-CF-Lua.

---

> > ### Author Response · Authors · 2025-12-03
> >
> > ### KL term and partial-credit ablations.
> > We further extended the "Training and Results" appendix.
> > To show how the models learned to abuse the partial-credit reward,
> > we evaluate their pass@1 rates on the test split of Ag-Codeforces-X.
> >
> > | Model           | Pass@1 |
> > |-----------------|--------|
> > | Qwen3 4B        | 0.0590 |
> > | Qwen3-4B-CF-Lua | 0.2476 |
> > | partial-r1      | 0.1333 |
> > | partial-r2      | 0.1562 |
> >
> > The models rewarded for partially-correct programs score far below
> > the model only given full-credit rewards.
> >
> > The dataset makes the partial-r1 and partial-r2 models more likely to
> > try abusing the metric like they did during training,
> > as the problems used to train these models come from the same dataset.
> > Meanwhile, the pass@1 metric only rewards programs which pass all tests.
> >
> > Regarding the KL term, we further note that
> > [Yue et al. (2025)](http://arxiv.org/abs/2504.13837)
> > also show an experiment where removing the KL term is beneficial
> > (see Section 4.4).
> >
> > ### Temperature and group size sweeps.
> > For convenience, we copy the result table we discussed in the previous comment.
> >
> > | Model           | Group size | Temperature | Pass@1 |
> > |-----------------|------------|-------------|--------|
> > | Qwen3-4B-CF-Lua | 32         | 0.7         | 0.2300 |
> > | normal-r1       | 32         | 0.7         | 0.1987 |
> > | normal-r2       | 32         | 0.7         | 0.2158 |
> > | size16-r1       | 16         | 0.7         | 0.1980 |
> > | size16-r2       | 16         | 0.7         | 0.1940 |
> > | size16-r3       | 16         | 0.7         | 0.1965 |
> > | size16-r4       | 16         | 0.7         | 0.2061 |
> > | size64-r1       | 64         | 0.7         | 0.2171 |
> > | size64-r2       | 64         | 0.7         | 0.2121 |
> > | size64-r3       | 64         | 0.7         | 0.2087 |
> > | temp0p2-r1      | 32         | 0.2         | 0.1991 |
> > | temp0p2-r2      | 32         | 0.2         | 0.2039 |
> > | temp0p2-r3      | 32         | 0.2         | 0.2198 |
> > | temp1-r1        | 32         | 1.0         | 0.2122 |
> > | temp1-r2        | 32         | 1.0         | 0.2148 |
> > | temp1-r3        | 32         | 1.0         | 0.2030 |

---

### Meta-Review · Area_Chair_4tu6 · 2026-01-09

**Summary:**

The authors proposed Agnostics, a post-training RL approach that is agnostic to programming languages. They proposed to use a single verifier to test solutions written in any language by only judging code solely by its externally observable behavior. Specifically, the authors introduce a new unit-test datasets with I/O format and use a verifier to judge and obtain verifiable rewards. The authors applied their methods in low-resource languages (Lua, Julia, R, etc.) and found improvements in different LLM families.

**Reviewer Concerns:**

- There is a concern that the approach is engineering-focused with less scientific contribution: the authors addressed this concern during rebuttal by emphasizing the importance of addressing harder coding tasks in low-resource languages.  I believed the contribution of this work is quite important and went beyond just an engineering trick.

- Lack of detailed ablation study e.g. KL term, partial credit rewards, temperature/group size, etc. : the authors added more results of ablation experiments during the rebuttal and I think they addressed this concern

- The number of languages being studied is limited to 5 and each of them is applied separately in training. A reviewer questioned whether there will be any negative transfer across programming languages. In the rebuttal, the authors show that there is no significant negative transfers across diverse programming languages.

- The model is limited to 8B

**Reviewer Scores:**

- Reviewer ZVFQ would increase their score from 4 to 5 given their concerns are sufficiently addressed
- Reviewer oRWA might keep the same score of 4 as some concerns were not addressed
- Reviewer pRJF would keep the same positive score of 8

---

### Decision · Program_Chairs · 2026-01-26

Accept (Poster)